# Vomeronasal Receptors Associated with Circulating Estrogen Processing Chemosensory Cues in Semi-Aquatic Mammals

**DOI:** 10.3390/ijms241310724

**Published:** 2023-06-27

**Authors:** Wenqian Xie, Meiqi Chen, Yuyao Shen, Yuning Liu, Haolin Zhang, Qiang Weng

**Affiliations:** Laboratory of Animal Physiology, College of Biological Sciences and Technology, Beijing Forestry University, Beijing 100083, China; xwq11@bjfu.edu.cn (W.X.); chenmq267@bjfu.edu.cn (M.C.); shenyuyao2021@sibcb.ac.cn (Y.S.); yuningliu9@bjfu.edu.cn (Y.L.)

**Keywords:** chemosensory processing, estrogen, semi-aquatic mammals, vomeronasal organ, vomeronasal receptors

## Abstract

In numerous animals, one essential chemosensory organ that detects chemical signals is the vomeronasal organ (VNO), which is involved in species-specific behaviors, including social and sexual behaviors. The purpose of this study is to investigate the mechanism underlying the processing of chemosensory cues in semi-aquatic mammals using muskrats as the animal model. Muskrat (*Ondatra zibethicus*) has a sensitive VNO system that activates seasonal breeding behaviors through receiving specific substances, including pheromones and hormones. Vomeronasal organ receptor type 1 (V1R) and type 2 (V2R) and estrogen receptor α and β (ERα and ERβ) were found in sensory epithelial cells, non-sensory epithelial cells and lamina propria cells of the female muskrats’ VNO. V2R and ERα mRNA levels in the VNO during the breeding period declined sharply, in comparison to those during the non-breeding period, while V1R and ERβ mRNA levels were detected reversely. Additionally, transcriptomic study in the VNO identified that differently expressed genes might be related to estrogen signal and metabolic pathways. These findings suggested that the seasonal structural and functional changes in the VNO of female muskrats with different reproductive status and estrogen was regulated through binding to ERα and ERβ in the female muskrats’ VNO.

## 1. Introduction

Information exchange can transmit information between different animals, which is essential for animal social behavior and physiology involving sexual recognition, individual survival and mating [1,2,3]. Chemical communication is an essential way of information exchange between individuals, which is usually less influenced by external environmental factors [4,5]. In mammalians, the olfactory and vomeronasal systems can receive a chemosensory signal that mediates their particular pattern of behaviors [6]. The main olfactory system is known to be closely related to the limbic system and has a crucial role in emotions and behavior. The vomeronasal system (VNS), considered an accessory olfactory system, which is associated with specific behaviors and hormonal responses to pheromone stimuli, is related to reproductive behavior, for instance, sexual attraction, maternal behavior or aggression [7,8,9]. The occurrence and development of the vomeronasal system varies greatly among mammals, depending on the environment in which they live [10]. Most rodents and land mammals have a VNS, but marine mammals and birds’ VNS disappeared [11]. It would appear that the vomeronasal system is of less use for airborne vertebrates and marine mammals [12]. In most mammals, the vomeronasal organ (VNO) comprises three parts: ducts, tissue and cartilage [13,14]. The vomeronasal duct (VND) is a bilateral mucus-filled superior duct at the bottom of the nasal septum and is typically crescent shaped with a thick sensory epithelium (SE) in the middle and ciliated non-sensory epithelium (NSE) on the sides [15]. The VND is surrounded by cartilage and parenchyma, and it consists of blood vessels, vomeronasal glands and connective tissue. In general, the VNO can pump liquids as well as gases into and out of the VND, thus enabling organisms to detect and distinguish between different compounds or odors, including volatile and water-soluble chemicals [16]. Therefore, the basic structure of the vomeronasal system is essential for intra- and inter-species information exchange.

Vomeronasal receptors (VRs) include two superfamilies, VRs type 1 and 2 (V1Rs and V2Rs). Both receptors have a seven-transmembrane region and are a G-protein-coupled receptor (GPCR) family [12]. However, V1Rs and V2Rs have numerous differences in properties that appear in their different functions [17]. The vomeronasal epithelium’s apical layer expresses V1Rs with Gαi2-coupled protein, whereas the vomeronasal epithelium’s basal layer expresses V2Rs with Gαo-coupled protein [11,18]. V1Rs are a large family of ~180 GPCRs encoded by one exon, which contains short extracellular domains at the N-terminal. Small volatile compounds implicated in gender discrimination can be found using V1Rs, which are closely connected to this [19], whereas V2Rs have a long extracellular region at the N-terminal, which can bind to water-soluble molecules involved in aggression, and the complete protein is encoded by six exons [20]. At the gene evolution level, during the vertebrate shift from aquatic to terrestrial life, the V2R repertoire contracts and the V1R repertoire expands to detect ligands in the air [11]. Recently, expression of the V1R gene has been found in rabbit, mouse, rat, dog, elephant and some primates [21,22,23]. However, the expression of the V2R gene has been described only in mouse, rat, rabbit and zebrafish [18,24]. The pattern of VRs expressed in species has been substantiated with the evolutionary history of vertebrates, which overlapped with a change in habitat from land to water [12]. Consistently, the active peptide ligands that bind to VRs usually also have an evolutionary transition from water to land. In mammals, steroids have been described as the crucial characterized source of VR ligands, and the VNO has been provided as the target of steroids [25,26]. After exposure to sex steroids, brains related to heterosexual and homosexual individuals’ mating and sexual behavior were found to be different than normal, demonstrating that sex steroids probably have a strong function related to mating or sexual behavior [27]. Previous studies reported that multiple sulfated steroids have been detected in mice as active compounds in nasal sensory neurons (VSNs), suggesting that steroid hormones are identified as an essential signaling characterization for chemical communication among mice [25,28]. Therefore, the VNO was the main target of sex steroids, which can identify various molecules in the environment that simultaneously regulate reproduction-related behaviors.

Large, semi-aquatic rodents known as muskrats were originally found in North America and have been introduced to Europe, Russia, China and Japan [29]. Muskrats are a robust vole with an average weight of about 1.5 kg and have a sparse scaly flat tail as long as the body which can act as a rudder in the water. Muskrats prefer to live in wetlands, marshes, lakes and streams. Muskrats have their name from glands in their perineum that produce a musky substance secreted into their urine through glandular ducts and used to mark their territory and communicate information [30]. As a typical seasonal animal, muskrats have active breeding from March to October. They are usually monogamous, and males secrete large amounts of odorous substances, mainly lipids and volatile substances, during the sexually active period [31]. Several studies have found that concentrations of sex steroid hormones and gonadotropic hormones significantly increased in May (the breeding period) and decreased in December (the non-breeding season) in female muskrats, suggesting that sex-related hormones might contribute to seasonal variation in the reproductive function of female muskrats [32]. Earlier research has documented that exocrine gland-secreting peptide 1 (ESP1), considered as a pheromone, can regulate female reproductive behavior via a specific vomeronasal receptor, suggesting that pheromones secreted by males can influence female reproductive behavior [33]. How female muskrats receive the substance secreted by the scent glands of male muskrats remains unclear. Our study investigates the expression patterns of VRs and estrogen receptors (ERs) in the female muskrats’ VNO to reveal that the morphological changes of the VNO associated with circulating estrogen could regulate female muskrat response to different chemical signals and individual behaviors in different reproductive statuses, which might play a significant part in the physiological foundation of semi-aquatic mammals’ response to chemosensory cues.

## 2. Results

### 2.1. Morphological Observations and Histological Features of the VNO of Female Muskrats

Morphological observations and histological features of the female muskrats’ VNO are presented in Figure 1. The VNO of female muskrats was observed as a paired tubular organ, which is situated at the basal part of the nasal septum (NS) (Figure 1A,B). Therefore, odor molecules are received through the nasal or oral cavity by the VNO (Figure 1B). Histologically, the VNO of the female muskrat is crescent-shaped and surrounded by the vomeronasal bone (Vo) on the periphery and separated by the NS in the middle (Figure 1C). The VNO consists of a vomeronasal duct (VND) and the lamina propria, including a vomeronasal gland (VG), blood vessel (BV), smooth muscles and connective tissue (Figure 1C). In both periods, the vomeronasal epithelium is separated into the sensory epithelium (SE) at the protruding part of the VND and the non-sensory epithelium (NSE) at the concave part of the VND (Figure 1D,E). The epithelium of the VNO consists of three parts: long columnar ciliated supporting cells (SC), sensory cells (SC) and basal cells (BC) near the lamina propria (LP) (Figure 1D,E). The ratio of surface epithelium to total NSE was calculated and is shown in Figure 1F. Based on the boxplot, the proportional contribution of the surface epithelial thickness was significantly higher in May than in December.

### 2.2. Immunohistochemical Locations of V1R, V2R, ERα and ERβ in the VNO of Muskrats

The immunohistochemical localizations of V1R, V2R, ERα and ERβ in the VNO of female muskrats are depicted in Figure 2. The immunoreactivity of V1R and V2R was localized mainly in the cytoplasm of the SE and NSE in the VNO in May and December (Figure 2A–D). The semi-quantitative positive staining of V1R was remarkedly higher in December than that of May (Figure 3A, Table 1), while the semi-quantitative positive staining of V2R was remarkedly higher in May than that in December (Figure 2C,D and Figure 3A, Table 1). The immunoreactivity of ERα and ERβ were localized mainly in the nucleus of the SE and NSE of the VNO in May and December (Figure 2I–L). The positive staining of ERα in May was significantly higher than that in December (Figure 2I,J and Figure 3A, Table 1). Meanwhile, relatively high positive staining for ERβ was observed in December (Figure 2K,L and Figure 3A, Table 1). In the negative controls, there was no signal visible (Figure 2E–H,M–P).

### 2.3. The mRNA Expressions of v1r, v2r, erα and erβ in the VNO of the Muskrats

The relative mRNA expression levels of *v1r*, *v2r*, *erα* and *erβ* were examined by real-time quantitative (qRT-PCR) (Figure 3B). The levels of *v1r* and *erβ* transcripts in the VNO of female muskrats declined remarkedly from December to May (*p* < 0.01) (Figure 3B). However, the mRNA expression levels of *v2r* and *erα* in the VNO of female muskrats increased significantly from May to December (*p* < 0.01) (Figure 3B). The ratios of VRs and ERs were calculated according to the mRNA expression levels in the VNO of female muskrats (Figure 3C,D). The ratio of V1R to V2R was notably higher in December compared to May (*p* < 0.01) (Figure 3C). However, the ratio of ERα to ERβ was higher in May than that in December (*p* < 0.01) (Figure 3D).

### 2.4. Concentration Profiles of 17β-Estradiol in the Female Muskrats

Female muskrats’ circulating levels of 17β-estradiol during both seasons were measured using ELISA (Figure 3E). The circulating 17β-estradiol concentration level in female muskrats reached the maximum in May and dropped remarkedly in December (*p* < 0.01).

### 2.5. Transcriptome Data Characterization

All transcriptome experiments were performed on both periods according to the data using sequencing with Illumina. Figure 4 displays the findings of the analysis of the transcriptome data in female muskrats’ VNO. A total of 18,339 gene fragments were detected to be co-expressed during both seasons. Respectively, 689 and 967 gene fragments were observed in May and December (Figure 4A). The volcano plot depicted that 883 differentially expressed genes (DEGs) were detected in the VNO, of which 554 genes were down-regulated and 329 genes were up-regulated (Figure 4B). The results of the DEG enrichment analysis are demonstrated in Figure 4C–F. The significantly differentially expressed pathways consisted of epithelial tube morphogenesis, mesenchyme development, regulation of epithelial proliferation and gland development, which includes some epidermal growth factors, such as the Sox family, *fgfr2*, *epha2* and *snal2*; these were identified via GO enrichment analysis and are shown in the chord diagram (Figure 4C). Bubble plots for the top 10 GO-enriched pathways are shown in Figure 4D. Consistently, GO enrichment analysis of all DEGs revealed that they are composed of the epidermal morphology pathway and gland development pathway (Figure 4D). There were six pathways related to odorant receptors: olfactory receptor activity, sensory perception of smell, odorant binding, pheromone receptor activity, response to stimulus and detection of chemical stimulus involved in sensory perception of smell, identified via cluster analysis of DEGs. The cluster of up-regulated genes comprised *aphrodisin*, *v1r*, *v2r* and *obp2b*, and down-regulated genes included *or10h2*, *lcn3* and *vn1r17p* (Figure 4E). Through performing KEGG enrichment analysis of DEGs, enriched pathways included the metabolic pathway, steroid hormone biosynthesis and the estrogen signaling pathway. Olfactory transduction is demonstrated in Figure 4F, implying that the steroid hormone and metabolic pathways may be involved in the regulation of seasonal differential expression of vomeronasal organs in female muskrats.

### 2.6. Molecular Docking Stimulation

The top five compounds detected via liquid and gas chromatography–mass spectrometry (LC-MS and GC-MS) in male muskrats’ scent glands are listed in Table 2 as previously reported [34,35,36,37]. In brief, the top five compounds with the highest concentration identified via the GC-MS method were selected. In the LC-MS method, the top five hormone-related compounds with the highest concentration were selected. The binding affinities and interactions between the selected molecular candidates and the V1R and V2R main protease are shown in Figure 5. The compounds’ binding affinities were scored using AutoDock Vina (1.1.2 for Windows, San Diego, CA, USA), and the value with highest negative is shown in Table 3. The highest negative values were selected after calculating the binding affinities. Among them, all binding ability was less than −4.0 KJ/mol; the binding ability of V2R and Androsterone sulfate was the strongest, and the binding affinity was −8.1 KJ/mol within interacting residues SER-84, THR-83 and GLU-14 (Table 3). Meanwhile, the binding ability of V2R and Muscone was −7.3 KJ/mol within interacting residues TYP-57. The binding ability of V1R, Muscone and Androsterone sulfate were −5.5 KJ/mol and −6.6 KJ/mol within interacting residues GLU-394 and GLN-217, respectively (Table 3).

## 3. Discussion

In mammals, due to multiple behavioral and reproductive strategies, the morphological characteristics of the VNO as well as the reaction to pheromones in the VNS are different in different species [2,38,39]. The VNO of hedgehogs consists of hyaline cartilage, a capsule and epithelial lumen soft tissue with an incisive duct directly into the nasal and oral epithelium, suggesting that odorants are mainly received through the nasal and oral cavity in the VNO of hedgehogs [14]. The brown bear’s VNO comprises the cartilage vomeronasal sensory epithelia (VNE), NSE, VG and soft tissue including the opening of the vomeronasal lumen into the mouth, raising the possibility that bears employ the VNO to detect pheromones in the environment [13]. In giraffes, the SE and NSE were described in the lateral and medial sections of the VNO lumen, which is covered by veins and numerous thin-walled vessels, indicating that the VNO has a potential role in regulating female giraffes’ function [40]. On the other hand, it is reported that individual status may affect the morphological characteristics of the vomeronasal epithelium [38,41]. For example, in rats, sex and age influence the morphological characteristics of the VNE [42]. In black bears, the mean thickness of VNE shows an age dependence, suggesting that VNO morphology may be affected by age and gender [13]. Our morphological characteristics and histological results indicated that the female muskrat’s VNO was well-developed, indicating that this organ was structurally able to receive external pheromones through the nasal or oral cavity, like most mammals. Consistently, the proportion of vomeronasal epithelium in female muskrats’ VNO in May was higher than that in December. Meanwhile, transcriptomic GO analysis depicted that DEGs were enriched into the epithelial morphological differential pathway. Therefore, the current results provided more evidence that seasonal variation of VNO morphology and histology in the female muskrat might closely be mediated by external chemical signals.

It is known that the VNO mainly binds different molecular chemicals through VRs to efficiently participate in response to the sensory environment as well as mediate different behavioral responses, for example, aggression and conspecific attraction [12,43]. Previous studies showed that odor directly induced changes in the expression of neurons and activated corresponding neuronal activation in the VNO epithelium, suggesting that the mammalian VNS can activate ligands of different VRs to detect odors secreted by other individuals [2,44]. Our results identified that the immunohistochemical expressions as well as gene expressions of V1R and V2R were shown in the female muskrats’ VNO, indicating that V1R and V2R might regulate the functions of the VNO in female muskrats. The immunoreactivity and transcriptional levels of V2R were notably higher in May compared to December. However, the expression levels of V1R were higher in December. Consistently, different chemical molecules had different binding abilities with V1R and V2R through the molecular docking analysis. The liquid and gas phase molecules androsterone sulfate and muscone had strong binding ability with V2R and V1R, respectively, indicating that V1R and V2R bind different ligands to respond to different environments. Differences in the structures and functions of V1R and V2R had also been found in other animals [12,45]. According to previous studies, differences in the expression of V1R and V2R in vertebrates may be related to the environment in which the organism lives [46,47]. In semiaquatic beavers, the V1R and V2R genes were described according to gene repertory analysis, suggesting that volatile and non-volatile substances can be received by beavers using the vomeronasal system [44]. In the rat and mouse, V1R and V2R counts were dynamic during different developmental periods, indicating that VRs might fulfill different behavior-related functions within the developmental process in rodents [48]. Several V2R genes have been found in both frogs and red-legged salamanders, while V1R repertoires were reduced, demonstrating that amphibians are well suited to terrestrial and aquatic habitats and could use water-soluble and volatile substances as chemical cues via various VRs [49,50]. Our results that the increase of V2R was expressed in the female muskrats’ VNO during the breeding period indicate that V2R is suspected of acting as the domain subtype VRs in the female muskrats’ VNO and playing a crucial role in chemical communication, including the recognition of water-soluble molecules and chemicals floating on the water surface. Thus, differential expression patterns of V1R and V2R during both periods in the female muskrats’ VNO might be consistent with the amphibious living environment of the muskrats via binding different molecular ligands responsible for essential functions.

In mammals, the VNO is regarded as a sensory organ and has been reported to be highly sensitive to steroid pheromones, which would be involved in mating and sexual behavior [8,51]. Recently, several studies found through immunohistochemical staining that mRNA levels of steroid hormone transport proteins and steroid hormone receptors are expressed in the SE of rodents’ VNO, indicating that steroid hormones projected the limbic system through the VNO [25,52]. In terrestrial salamanders, circulating steroid hormone levels were found to stimulate the morphological characteristic of chemosensory cues in the VNO during the mating season [53]. In this study, ERs were colocalized with VRs in the NSE and SE cells of the VNO, indicating possible cross talk between the estrogen signaling pathway and chemosensory cue processing. Circulating peptides or steroid hormones could modulate odorant responses to alter the behavior of animals [25,54]. In particular, 17β-estradiol may be synthesized locally in the VNO to readily regulate the VSN-mediated odor response [25]. Although the expression pattern of factors related to the estrogen signaling pathway was not completely consistent with VRs, it cannot be excluded that estrogen may be involved in regulating VR expression through modulating the expression of different types of ERs. Furthermore, molecular docking results show that V1R and V2R have different affinities for different ligands. It has been reported that estradiol may mediate the action of the VNO through reducing the response of calcium ions in urine to steroid sulfate. In the most recent research, nuclear and membrane estrogen receptors can synergistically regulate estrogen’s numerous functions [55]. In addition to its regulatory role through nuclear estrogen receptors, estrogen can also rapidly regulate estrogenic signaling through membrane estrogen receptors, particularly in the nervous system. Among them, membrane-associated estrogens α and β, as well as GPER1 and ER-X, can act through the second messengers, thus acutely regulating estrogenic signaling throughout the organism [55]. Rapid responses to external stimuli via membrane estrogen receptors can initiate the transcription of diverse estrogen-regulated genes. The role of estrogen receptors in the vomeronasal organ of female muskrats may also be through the reception of membrane estrogen receptors, resulting in rapid signaling and reception in the vomeronasal organ. Therefore, using the semi-aquatic muskrat as a model, our results elucidate that circulating estrogen may cooperatively regulate the differential expression of VRs through the locally expressed ERs, thus affecting chemosensory processing in the VNO. Together, the results indicate that the female muskrats’ VNO was a main target of estrogen, and estrogen might have a vital role in pheromone-dependent behavioral alterations through regulating the VRs and ERs, including more activity in receiving chemical signals to prepare for reproduction.

## 4. Materials and Methods

### 4.1. Animals

The Beijing Forestry University Experimental Animal Ethics Committee gave its approval to all animal experiment procedures. All female muskrats were obtained in May (the breeding, B, *n* = 12) and December (the non-breeding season, NB, *n* = 12) from Xinji Muskrats Farm in Hebei Province, China. The heads were quickly dissected from female muskrats after being paralyzed using diethyl ether. The samples were promptly frozen at −80 °C for the transcriptome analysis and the molecular experiments; other tissues were fixed in paraformaldehyde for 24 h and then switched to 70% alcohol for the following procedures. For the subsequent hormonal study, serum was isolated from blood by means of centrifuging at 3000× *g* for 20 min. The animal study protocol was approved by the Experimental Animal Ethics Committee of Beijing Forestry University. Approval Code EAWC_BJFU_2021005. Approval Date: 11 March 2021.

### 4.2. Macroscopic Anatomy and Histological Process

One of the female muskrats’ heads was dissected freshly for macroscopic anatomical evaluation of the VNO prior to the histological process outlined below. In accordance with the accepted procedure, the VNO was taken out of the nasal cavity and put into paraffin. In brief, the samples were calcified and dehydrated in succession of alcohol and xylene series and embedded in paraffin wax. The samples were subsequently sliced into transverse sections that were 5 μm thick for the general histological and immunohistochemical observations. Hematoxylin-eosin (H&E) staining was used to perform histological analysis in some sections.

### 4.3. Immunohistochemical Process

To lessen background staining brought on by the second antibody, 10% normal goat serum was incubated with the serial slices of the VNO tissues. Then, the primary antibody (1:200 dilution) against VMN1R41 (ARP96665_P050, Aviva Systems Biology, San Diego, CA, USA), VN2R1P (OASG07552, Aviva Systems Biology, San Diego, CA, USA), AR (sc-7305, Santa Cruz Biotechnology, Santa Cruz, CA, USA), ERα (sc-542, Santa Cruz Biotechnology, Santa Cruz, CA, USA), ERβ (sc-390243, Santa Cruz Biotechnology, Santa Cruz, CA, USA) for 12 h at 4 °C. Instead of using the main antibodies, normal rabbit IgG was applied at a ratio of 1:10000 to the control sections. All sections were incubated with a second antibody, goat anti-mouse or goat anti-rabbit IgG, conjugated using a Kit as we described before, after which the coloring was observed using DAB as the chromogen. The slides incubated with anti-VMN1R41 and anti-VN2R1P were counterstained with hematoxylin solution. Analysis was performed using a relative quantitative positivity score (ranging from negative − to very strong positive +++) [31]. The positive optical density of all cells stained with target antibodies in the muskrat VNOs was assessed and analyzed with Fiji software (version 2.13.1) and GraphPad Prism 8.0 (GraphPad Software Inc., San Diego, CA, USA).

### 4.4. RNA Isolation and Quantitative Real-Time Polymerase Chain Reaction (qPCR)

Following the manufacturer’s instructions, total RNA was extracted from each sample using the TRIzol reagent (Invitrogen Co., Carlsbad, CA, USA). A reverse transcription kit (Promega Corporation, Madison, WI, USA) was used to synthesize the first-strand cDNA from total RNA. The amplification primer sequences for qPCR were summarized in Table 4. The qPCR reaction conditions were as follows: preheating for 10 min at 95 °C, followed by 40 cycles of 30 s at 95 °C, 30 s at 60 °C, 30 s at 72 °C and ending with melting curves. Each sample were conducted in triplicate in individual experiments. The target and reference genes’ PCR efficiency was similar, and the intra-assay variation was less than 10%. Using the 2^−ΔΔCq^ method, the relative expression level of each target mRNA in relation to *β-actin* was determined.

### 4.5. Hormone Assay

The female muskrats’ serum was centrifuged for 15 min at 1000× *g* at 4 °C. Using ELISA Kit (CSB-E05109m for 17β-estradiol, Cusabio Biotech Co., Ltd., Wuhan, China), the concentrations of 17β-estradiol were determined. Finally, a microplate reader (PT 3502G, Beijing Potenov Technology Co., Ltd., Beijing, China) was used to color the assay plate and read it at 450 nm within 10 min. Each sample was measured in duplicate. The parallelism between the standard curve and the series diluted sample curve was examined in order to establish the validity. The assay’s sensitivity was 40 pg/mL. The co-efficients of variation for the intra- and inter-assays were 1.25% and 3.87%, respectively. 

### 4.6. Library Preparation for Transcriptome Sequencing

The RNA sample preparations employed a total of 1.5 g RNA per samples (*n* = 3, each time). The manufacturer’s proceeding states that the NEBNext^®^ UltraTM RNA Library Prep Kit for Illumina^®^ (NEB, Ipswich, MA, USA) was used to create the RNA-sequencing libraries. Using poly-T Oligo-attached (dT) magnetic beads, the mRNA was extracted and enriched from total RNA. M-MuLV Reverse Transcriptase (RNase II), DNA Polymerase I and random hexamer primer were used to create the double-stranded cDNA from the RNA. The AMPure XP beads were used to purify the double-stranded cDNA, and then PCR amplification was used to create a cDNA library for later library detection and sequencing via synthesis.

### 4.7. Transcriptome Analysis

Adapters and low-quality reads were removed from raw reads to get clean data. Using the previously described method, the measurement of gene expression level was obtained [56]. Using the DESeq R package, the differentially expressed genes (DEGs) of two periods were examined and identified. The genes that were differentially expressed were identified using an adjusted *p*-value of 0.05. Further implementation of enrichment analyses of DEGs using Gene Ontology (GO) and Kyoto Encyclopedia of Genes and Genomes (KEGG) was performed with KOBAS software (version 3.0, accessed on 1 January 2023) [57].

### 4.8. Protein-Ligand Interaction Analysis

Autodock Vina (1.1.2 for Windows, San Diego, CA, USA) was employed to analyze the binding affinities and modes of interaction between the chemicals and vomeronasal receptors [58]. The molecular structures of volatile chemicals and water-soluble chemicals (androsterone sulfate compound CID 159663 and muscone compound CID 10947) were required from PubChem Compound (https://pubchem.ncbi.nlm.nih.gov/, accessed on 1 January 2023). Using Swiss-model software (https://swissmodel.expasy.org/, accessed on 1 January 2023), 7dd6.1.A and 7l0p.1.A were used as models to predict the tertiary structure of V1R and V2R in the female muskrats’ VNO, and the sequence with high similarity was selected for the next analysis. After removing all water molecules and adding polar hydrogen atoms, the protein and ligand files were transformed into the pdbqt format. To support free molecular motion and cover the structure of each protein, a cubic grid box with dimensions of 25 × 25 × 34 Å was constructed. Protein–ligand interaction analysis was carried out using AutoDock Vina (1.1.2 for Windows, San Diego, CA, USA), and model visualization and analysis were performed using PyMol (version 2.5.0) [59].

### 4.9. Morphometry and Statistical Analysis

Using the arbitrary distance approach, the thickness of the NSE was measured in micrometers (μm) from the basement membrane to the surface. All measurements were observed under a 10× objective lens with an H&E-stained section, and each measurement was taken 10 times to calculate the relative mean value of NSE thickness. GraphPad Prism 8.0 was used for the statistical analyses after the trial produced quantitative data. To assess group differences, Student’s *t*-test was employed. Statistical values of *p* < 0.05 were regarded as significant.

## 5. Conclusions

To date, this study is the first report to elucidate the morphological and functional alterations of the VNO associated with various reproductive statuses in semi-aquatic mammals. This study highlighted that the female muskrats’ VNO was the target organ for estrogen, and estrogen might regulate the female muskrats’ VNO response to chemical signals through ERs in coordination with VRs in May and December. Although the reason for the interaction mechanism of ERs and VRs in the VNO of female muskrats remains unclear, the present results suggest the VNO might be the primary target of estrogen in the response to volatile and water-soluble molecules via ERs and VRs to regulate their physiological role in different reproductive statuses. We will further identify the neuronal circuits regulated by natural pheromones coordinating with sex steroids in the brain, which can facilitate our understanding of chemical communication in mammals.

## Figures and Tables

**Figure 1 ijms-24-10724-f001:**
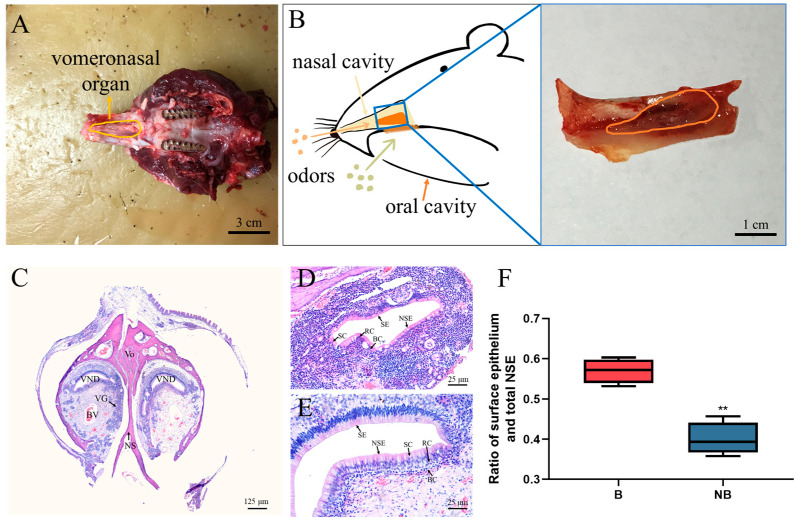
Morphological and histological features of the vomeronasal organ (VNO) in female muskrats. (**A**) The topography of female muskrats’ VNO; (**B**) schematic diagram of relative position of VNO; (**C**) sagittal section of the VNO showing its main microscopic features: vomeronasal bone (Vo), nasal septum (NS), vomeronasal duct (VND), vomeronasal gland (VG), blood vessel (BV); (**D**,**E**) transverse section of the VND showing its main microscopic features: sensory epithelium (SE), non-sensory epithelium (NSE), supporting cells (SC), sensory cells (SC) and basal cells (BC) during the breeding (B) and non-breeding seasons (NB), respectively; (**F**) ratio of surface epithelium to total NSE in the breeding and non-breeding seasons. Scale bars: 3 cm (**A**), 1 cm (**B**), 125 μm (**C**), 25 μm (**D**,**E**). The error bars represent means ± SEM (*n* = 6, each period). * Statistically significant values (** *p* < 0.01).

**Figure 2 ijms-24-10724-f002:**
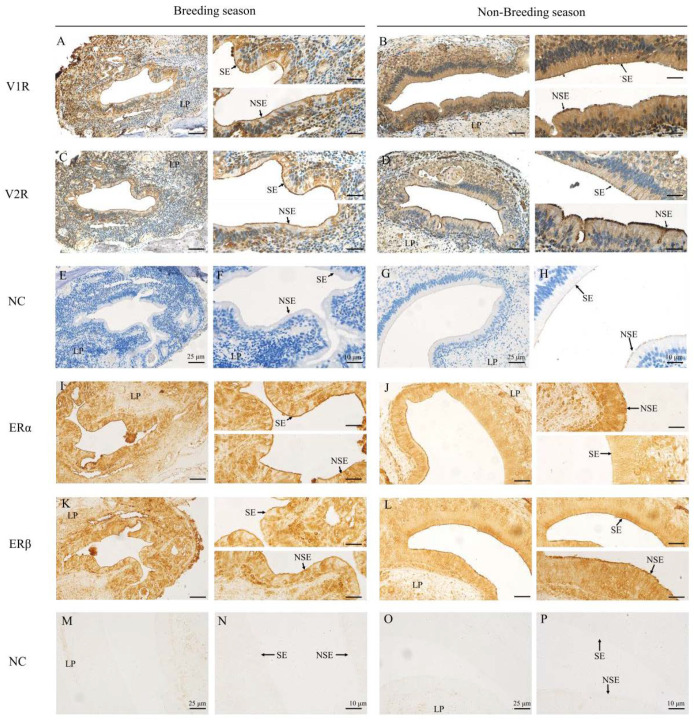
Vomeronasal receptor type-1 (V1R), vomeronasal receptor type-2 (V2R), estrogen receptor α (ERα) and estrogen receptor β (ERβ) immunolocalizations of the vomeronasal organ (VNO) in female muskrats. The immunolocalizations of V1R (**A**,**B**), V2R (**C**,**D**), ERα (**I**,**J**) and ERβ (**K**,**L**) in the VNO of the female muskrats during the breeding and non-breeding seasons. The negative control (NC) of the VNO of the female muskrats’ sections in the breeding season and the non-breeding season (**E**–**H**,**M**–**P**). The second and fourth columns of the figure show a zoom into the immunolocalizations. Scale bars: 25 μm (the first and third columns) and 10 μm (the second and fourth columns). LP, lamina propria; SE, sensory epithelium; NSE, non-sensory epithelium.

**Figure 3 ijms-24-10724-f003:**
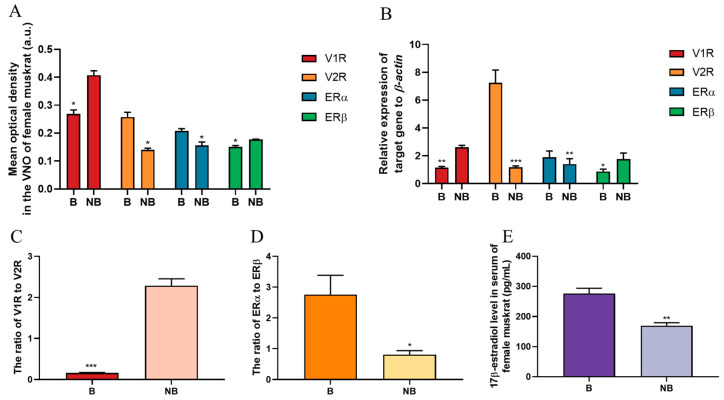
The distributions and expression levels of vomeronasal receptor type-1 (V1R), vomeronasal receptor type-2 (V2R), estrogen receptor α (ERα) and estrogen receptor β (ERβ) in the VNO of female muskrats during the breeding season and the non-breeding season. (**A**) The mean optical density of V1R (red), V2R (orange), ERα (blue) and ERβ (green) in the VNO of the female muskrats during different seasons. (**B**) The relative mRNA levels of V1R (red), V2R (orange), ERα (blue) and ERβ (green) in the VNO of female muskrats during different seasons. The ratio of V1R to V2R (**C**) and ERα to ERβ (**D**) in the VNO of female muskrats through an analysis of mRNA expression levels of VRs and ERs. (**E**) The concentration of 17β-estradiol in the serum of female muskrats during the breeding and non-breeding seasons. The error bars represent means ± SEM (*n* = 6, each period). B, the breeding season; NB, the non-breeding season. * Statistically significant values (* *p* < 0.05; ** *p* < 0.01; *** *p* < 0.001).

**Figure 4 ijms-24-10724-f004:**
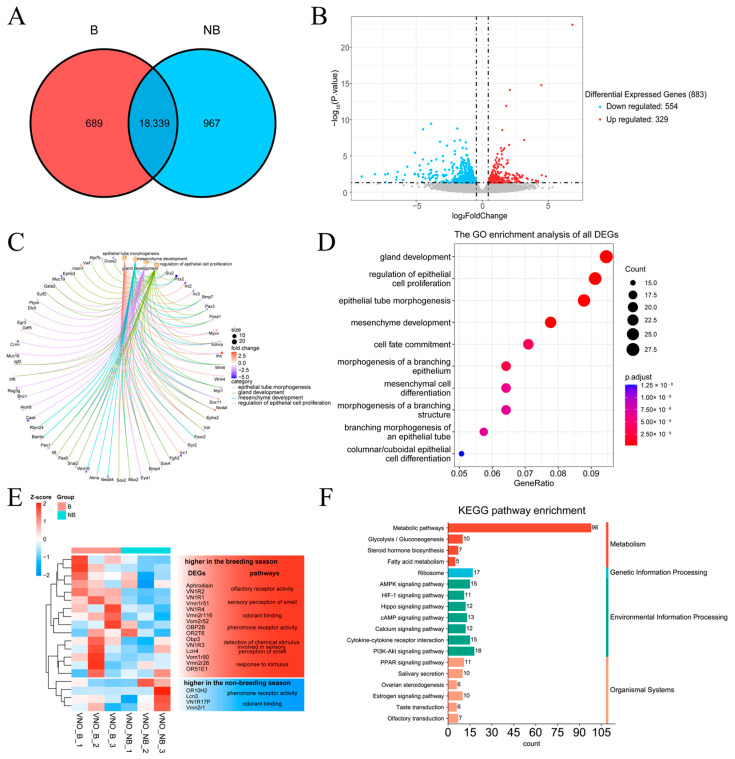
The transcriptome analysis of the vomeronasal organ (VNO) during the breeding and non-breeding seasons. (**A**) Venn diagram showing the shared and unique fragment between the two compared groups of the VNO. (**B**) Volcano plot displaying the up- and down-regulated genes in the VNO between the breeding and nonbreeding seasons. (**C**) Circle plot showing the GO pathway and differentially expressed genes (DEGs). (**D**) Bubble plot displaying the GO enrichment pathways of all DEGs. (**E**) Heatmap diagram illustrating the expression of up-regulating and down-regulating DEGs and pathways in the different seasons. (**F**) Bar diagram depicting the KEGG pathway enrichment of all DEGs in the VNO. B, the breeding season; NB, the non-breeding season.

**Figure 5 ijms-24-10724-f005:**
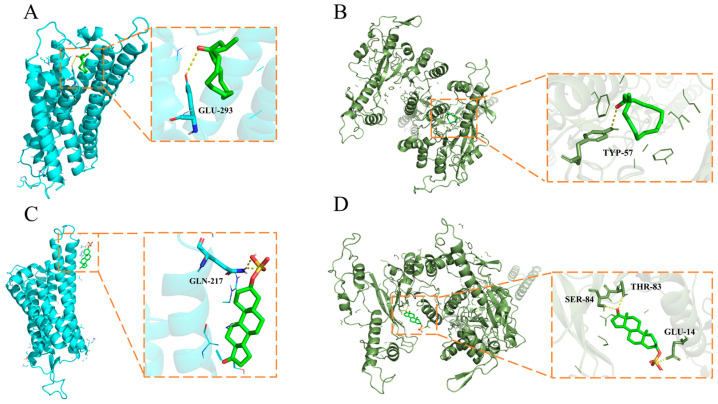
Ligand poses in vomeronasal receptor type 1 (V1R) and vomeronasal receptor type 2 (V2R) binding sites. (**A**) The location of ligand binding between muscone and V1R (blue). (**B**) The location of ligand binding between androsterone sulfate and V1R (blue). (**C**) The location of ligand binding between muscone and V2R (forest green). (**D**) The location of ligand binding between androsterone sulfate and V2R (forest green). The inset shows a zoom into the binding site and ligand residues.

**Table 1 ijms-24-10724-t001:** Immunoreactivity of V1R, V2R, ERα and ERβ in the VNO of female muskrats during the breeding and non-breeding seasons.

Antibodies	Breeding Season		Non-Breeding Season	
	SE	NSE	SE	NSE
V1R	++	++	+++	+++
V2R	+++	+++	++	++
ERα	+++	+++	+	++
ERβ	++	++	+++	+++

The immunohistochemical staining was determined as negative (−), positive (+), strongly positive (++) and very strongly positive (+++). Staining that was weak but higher than that of the control was set as positive (+). The highest intensity staining was set as very strongly positive (+++). A staining intensity between + and +++ was set as strongly positive (++). No signal was set as negative (−). SE means sensory epithelium; NSE means non-sensory epithelium.

**Table 2 ijms-24-10724-t002:** GC-MS and LC-MS analysis showed metabolite compounds in the scent glands of male muskrats.

Metabolite Compound	MS Platform	References
Muscone	GC-MS	[36,37]
Cyclopentadecanone	GC-MS	[34,35,37]
Cyclopentadecanol	GC-MS	[34,35,37]
Palmitoleic acid	GC-MS	[35]
Z-7-Hexadecenoic acid	GC-MS	[35]
PC (16:0/16:0)	LC-MS	[35]
Phosphatidylethanolamine	LC-MS	[35]
LysoPC (16:0)	LC-MS	[35]
Androsterone sulfate	LC-MS	[35]
Dopamine	LC-MS	[35]

**Table 3 ijms-24-10724-t003:** Docking results for the compound and protein.

Compound	Protein Name	Docking Score (KJ/mol)	Interacting Residues
Muscone	V1R	−5.5	GLU-394
Androsterone sulfate	V1R	−6.6	GLN-217
Muscone	V2R	−7.3	TYP-57
			SER-84
Androsterone sulfate	V2R	−8.1	THR-83
			GLU-14

**Table 4 ijms-24-10724-t004:** Primer sequences used for mRNA qRT-PCR.

Gene	Primer Sequence	Product Length (bp)	Accession Numbers
*v1r*	F: 5′-ATGCAGCACATCCTCACTCC-3′	205	NM_001167146.1
	R: 5′-AGATACTGGGGAAGACCGCA-3′		
*v2r*	F: 5′-AAGGAGCCAGTTCTCACTGC-3′	285	NM_001385059.1
	R: 5′-TACATTCCGCACTGCACACA-3′		
*erα*	F: 5′-CGACTATGCCTCTGGCTACC-2′	157	NM_012689.1
	R: 5′-TCATCATGCCCACTTCGTAA-2′		
*erβ*	F: 5′-CCTTGGTGTGAAGCAAGATC-3′	286	NM_012754.3
	R: 5′-GGATCCACACTTGACCATTC-3′		
*actb*	F: 5′-GACTCGTCGTACTCCTGCTT-3′	223	NM_007393.5
	R: 5′-AAGACCTCTATGCCAACACC-3′		

## Data Availability

Not applicable.

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
