# Peer review of "Vomeronasal Receptors Associated with Circulating Estrogen Processing Chemosensory Cues in Semi-Aquatic Mammals"

_ijms, 2023, doi:10.3390/ijms241310724_

Round 1
Author Response
Reviewer 1:
Response: Thank you very much for the reviewer’s comment. Revisions and changes are made accordingly, and please find them highlighted with red color in the revised manuscript.
Minor editorial revisions will be needed.
- Line 52: vomeronasal receptors should be VRs
Response:
Thank you for your comment. “vomeronasal receptors” has been changed to “VRs”. Please see Line 52.
- Line 133: add number of replicates as n=
Response:
Thank you for your comment. The number of replicates has been added. Please see Line 132-133.
- Line 190: Illumine should be Illumina ?
Response:
Thank you for your comment. “Illumine” has been changed to “Illumina”. Please see Line 190.
- Line 193: DEGs should be differentially expressed genes (DEGs)
Response:
Thank you for your comment. “DEGs” has been changed to “differentially expressed genes (DEGs)”. Please see Line 193-194.
- Lines 226, 409, 420: is the California city name?
Response:
Thank you for your comment. We have added the city name—“San Diego” before the CA. Please see Line 230, 421, 432.
- Line 355: delete San Diego, CA, USA
Response:
Thank you for your comment. “San Diego, CA, USA” has been deleted. Please see Line 368.
- Lines 357-358: delete Santa Cruz, CA, USA
Response:
Thank you for your comment. “Santa Cruz, CA, USA” has been deleted. Please see Line 369-370.
- Line 372: add city before CA
Response:
Thank you for your comment. The city name “Carlsbad” has been added before CA. Please see Line 384.
- Line 373: add WI before USA
Response:
Thank you for your comment. “WI” has been added before USA. Please see Line 385.
10.Line 389: delete % after 1.25
Response:
Thank you for your comment. We calculated the inter- and intra-assay coefficients of variability to assess the reproducibility of ELSIA. The coefficients of inter- and intra-assay in this study are 1.25% and 3.87%, respectively. The minimal requirement of inter- and intra- assay is less than 15%. Therefore, our data meet the criteria of coefficients of inter- and intra- assay. The symbol “%” should not be deleted.
11.Ref. no: 9, 12, 14, 16, 23, 34, 37, 45, 54: titles should not be used large capital
Response:
Thank you for your comment. The format of titles has been changed in the reference 9, 12, 14, 16, 23, 34, 37, 45, 54. Please see Line 480-481, 485-486, 489-490, 493-494, 507-509, 532-533, 538-539, 555-556, 572-573.
12.Lines 523, 525, 527: journal names should be appropriately abbreviated
Response:
Thank you for your comment. The abbreviation of journal names has been changed in the reference 35, 36, 37. Please see Line 535, 536-537, 539.
13.Lines 533, 551: scientific names should be italic
Response:
Thank you for your comment. The scientific names have been italic. Please see Line 545, 563.
14.Line 554: journal name should be italic
Response:
Thank you for your comment. The journal name has been italic. Please see Line 566.

Reviewer 2 Report
Dear Authors,
The article “Vomeronasal receptors associated with circulating estrogen processing
chemosensory cues in semi-aquatic mammals” is very interesting and modern, as it describes the association between vomeronasal system and sex hormones. This has not been mentioned before in the scientific literature. Different expression of the ER subtypes depends on the season, it introduce new insights in the mechanisms of steroid hormones action.
Some details and aspects are not written.
1) In the abstract VR and ER should be abbreviated
2) “KEGG enrichment analysis of DEGs, enriched pathways including metabolic pathway, steroid hormone biosynthesis and estrogen signaling pathway, olfactory transduction were demonstrated in Figure 4F”. What is the sense of this figure and what are the highlights of this analysis?
3.) In the table 3 it is better to show numbers as a references like [34,37]
4.) “The top five compound that detected by liquid and gas chromatography–mass spectrometry (LC-MS and GC-MS) in the male muskrats’ scent glands”. What the aspects dictated the choice of these 5 compounds? Based on their concentration or something else? And why Androsterone sulfate but not DHEA sulfate?
5.) It would be interesting to make a docking not only with Androsterone sulfate and Muscone, but also with other native ligands. Or it is desirable to explain the choice.
6.) Why female muskrats were investigated? In the previous article that is often cited here (Xie et al., 2022), males have been used.
7.) Some discussion is desirable about the role of membrane estrogen receptors in the chemosensoryprocesses
The highlights of this article would be helpful for the readers.
Author Response
Reviewer 2:
Dear Authors,
The article “Vomeronasal receptors associated with circulating estrogen processing
chemosensory cues in semi-aquatic mammals” is very interesting and modern, as it describes the association between vomeronasal system and sex hormones. This has not been mentioned before in the scientific literature. Different expression of the ER subtypes depends on the season, it introduce new insights in the mechanisms of steroid hormones action.
Some details and aspects are not written.
Response:
Thank you very much for the reviewer’s comment. Revisions and changes are made accordingly, and please find them highlighted with red color in the revised manuscript.
1) In the abstract VR and ER should be abbreviated
Response:
Thank you very much for the reviewer’s comment. The abbreviation of VR and ER has been changed in “abstract” section. Please see Line 15-16.
2) “KEGG enrichment analysis of DEGs, enriched pathways including metabolic pathway, steroid hormone biosynthesis and estrogen signaling pathway, olfactory transduction were demonstrated in Figure 4F”. What is the sense of this figure and what are the highlights of this analysis?
Response:
Thank you very much for the reviewer’s comment. These results suggest that these pathways may be involved in the regulation of seasonal differential expression of vomeronasal organs in the female muskrats. The DEGs were identified by transcriptomic analysis, but the specific functions of DEGs could be enriched by KEGG analysis. Therefore, the enrichment pathway of KEGG may have a regulatory effect on the seasonal changes of vomeronasal organ of female muskrat. The sense and highlights have been added in the “results” section. Please see Line 208-212.
3.) In the table 3 it is better to show numbers as a references like [34,37]
Response:
Thank you very much for the reviewer’s comment. The format of references shown in Table 3 has been changed to reference numbers like [34, 37]. Please see Table 3, Line 245-247.
4.) “The top five compound that detected by liquid and gas chromatography–mass spectrometry (LC-MS and GC-MS) in the male muskrats’ scent glands”. What the aspects dictated the choice of these 5 compounds? Based on their concentration or something else? And why Androsterone sulfate but not DHEA sulfate?
Response:
Thank you very much for the reviewer’s comment. We selected the top five compounds with highest concentration from the GC-MS data. In the LC-MS data, we selected the top five hormone-related compounds with highest concentration.
In LC-MS analysis, we compared the ion peak and residence time with database to identify secondary compounds, and identified androsterone sulfate which is one of the top five hormone-related compounds with highest concentration. In the same data, we did not find the no ion peak and residence time of DHEA sulfate. Therefore, we choose androsterone sulfate.
5.) It would be interesting to make a docking not only with Androsterone sulfate and Muscone, but also with other native ligands. Or it is desirable to explain the choice.
Response:
We appreciate the reviewer’s comments. Until now, native ligands of the vomeronasal receptors in the female muskrats are not well understood. The purpose of this article is to identify potential native ligands of the vomeronasal receptors in semi-aquatic animals. Most previous studies have found the odorous substance of scent glands contain pheromones, which can induce reproductive behavior of muskrats. We identified components in odorous substance of the scent gland by GC-MS and LC-MS data, including cycloalkanones, fatty acids and steroidal compounds. Those substance may be the potential native ligands of VRs.
First discovered in insects and later found in mammals, pheromones are chemicals that can be secreted outside of body through fluids such as urine and sweat. At the same time, pheromones can induce hormonal changes or certain behaviors. Based on the previous literature, we selected two different substances in the gas phase (muscone) and the liquid phase (androsterone sulfate) which was secreted outside through scent glands for molecule docking analysis to investigate the native ligand in the semi-aquatic mammals. The docking results found that androsterone sulfate and muscone have a strong binding ability with VRs and regarded as potentially important native ligands in the scent gland of male muskrats. The specific functions and binding sites of androsterone sulfate and muscone still need to be further studied and confirmed in structural biology. The reviewer proposed an excellent point, and we will further investigate the structure and function of the native ligands of the vomeronasal receptors to providing essential information regarding the semi-aquatic mammals’ chemosensory processes.
6.) Why female muskrats were investigated? In the previous article that is often cited here (Xie et al., 2022), males have been used.
Response:
Thank you very much for the reviewer’s comment. The scent gland is found only in male muskrats and is a unique organ of male muskrats. The odorous substance secreted by scent glands can regulate female reproductive behavior. As the typical seasonal breeder, the male muskrat secretes large amounts of odorous substances, including lipids and macromolecules during the reproductive phase. However, the mechanism of how the female muskrat receives the substance secreted by the scent glands of the male one within the environments during different seasons remains unclear. Therefore, our study mainly used the female muskrats’ vomeronasal organ as the subject. In this study, we characterized the functional olfactory receptor--vomeronasal receptors using immunohistochemical and molecular methods, transcriptome as well as molecular docking to elucidate the mechanism underlying chemical communication associated with reproductive behaviors in vertebrates. In this study, we investigated the potential molecules native ligands with VRs to elucidate the physiological foundation of semi-aquatic mammals in response to chemosensory cures through VRs.
7.) Some discussion is desirable about the role of membrane estrogen receptors in the chemosensory processes
Response:
Thank you very much for the reviewer’s comment. In the most recent researches, nuclear and membrane estrogen receptors can synergistically regulate estrogen numerous functions [1]. In addition to its regulatory role through nuclear estrogen receptors, estrogen can also rapidly regulate estrogenic signaling through membrane estrogen receptors, particularly in the nervous system. Among them, membrane-associated estrogens α and β, as well as GPER1, ER-X can act through the second messengers, thus acutely regulating estrogenic signaling throughout the organism [1]. Rapid responses to external stimuli via membrane estrogen receptors can initiate the transcription of diverse estrogen-regulated genes. The role of estrogen receptors in the vomeronasal organ of the female muskrats may also be through the reception of membrane estrogen receptors, resulting in rapid signaling and reception in the vomeronasal organ. The discussion about the role of membrane estrogen receptors in the chemosensory processes have been added in the “discussion” section. Please see Line 326-336 and Reference 56, Line 576-577.
[1] Wnuk, A.; Przepiórska, K.; Pietrzak, B.A.; Kajta, M. Emerging Evidence on Membrane Estrogen Receptors as Novel Therapeutic Targets for Central Nervous System Pathologies. Int. J. Mol. Sci. 2023, 24, 4043. https://doi.org/10.3390/ijms24044043
The highlights of this article would be helpful for the readers.
Response:
Thank you very much for the reviewer’s comment. The reviewer proposed an excellent point. Unfortunately, the journal does not have “highlights” sections. We have written the highlights as following below.
Highlights:
- Seasonal changes of morphology and histology were detected in the VNO of female muskrats.
- The expression levels of V1R, V2R, ERα and ERβ are higher in the breeding seasons.
- Liquid and gas phase molecules androsterone sulfate and muscone have binding abilities with VRs.
- Estrogen regulates the VNO’s function through binding to ERs in the female muskrats.
